# The Social Rank of Zoo-Housed Japanese Macaques is a Predictor of Visitor-Directed Aggression

**DOI:** 10.3390/ani9060316

**Published:** 2019-06-02

**Authors:** Jocelyn M. Woods, Stephen R. Ross, Katherine A. Cronin

**Affiliations:** 1Lester E. Fisher Center for the Study and Conservation of Apes, Lincoln Park Zoo, 2001 N. Clark Street, Chicago, IL 60614, USA; jocelyn.woods@czs.org (J.M.W.); kcronin@lpzoo.org (K.A.C.); 2Animal Welfare Research, Chicago Zoological Society, 3300 Golf Road, Brookfield, IL 60513, USA; 3Animal Welfare Science Program, Lincoln Park Zoo, 2001 N. Clark Street, Chicago, IL 60614, USA

**Keywords:** Elo-rating, animal welfare, visitor effects, hierarchy, primate behavior, zoo animal welfare

## Abstract

**Simple Summary:**

Millions of people visit zoological parks around the world every year. As a result, there has been significant interest in how the presence of these visitors affects the behavior and wellbeing of zoo animals, of which primates are a group that has been extensively studied in this regard. However, many studies fail to consider the possibility that group dynamics, particularly social rank, may affect an individual animal’s reaction to visitors. We investigated how visitor presence, both crowd size and activity level, along with the individual rank of zoo-housed monkeys (Japanese macaques) influenced the occurrence of behavior directed toward the visitors. Behaviors directed toward visitors were predominantly threatening. The results revealed that the social rank of a monkey was a significant predictor of visitor-directed aggression, and lower-ranked individuals displayed more frequent aggression toward visitors. We think it is likely that this finding is a result of low ranking individuals redirecting their aggression to visitors as safe targets. Furthermore, the activity level of the visitors affected visitor-directed aggression by the monkeys more so than the size of the crowd. These findings highlight that although environmental factors impact the behavior of animals at the zoo, social rank is also important to consider when understanding how animals may react to zoo visitors.

**Abstract:**

The effect that visitors have on the behavior and welfare of animals is a widely-studied topic in zoo animal welfare. Typically, these studies focus on how the presence or activity levels of visitors affect animals. However, for many species, and particularly primates, social factors, such as social rank, can also have a large impact on behavior. Here, we considered the influence of both the role of visitors (crowd size and activity levels) and rank on the occurrence of visitor-directed aggression by zoo-housed Japanese macaques (*Macaca fuscata*, N = 12). We conducted 52 weeks of observation (443.8 h) of macaques living in a large outdoor habitat and recorded 1574 events of visitor-directed behavior, 94.2% of which was characterized as aggressive. We calculated rank using the Elo-rating method. GLMM comparisons indicate that rank was a significant predictor of visitor-directed aggression, with lower-ranked individuals displaying more frequent aggression towards visitors. Additionally, visitor-directed aggression differed by crowd activity levels, but not crowd size. These results support our prediction that rank is associated with differences in visitor-directed aggression, and we interpret this pattern as lower-ranking macaques redirecting aggression toward zoo visitors as safe targets. This work emphasizes how factors emanating from the zoo environment can combine with social dynamics to influence primate response to human presence in the zoo setting.

## 1. Introduction

Understanding how zoo visitors impact the welfare of animals is essential to creating zoo environments in which animals thrive, yet the magnitude and direction of the reported effects of visitors on animal welfare have varied. While many studies have revealed negative impacts of visitors on animal welfare [1], there is also the potential for visitor presence to lead to neutral or positive welfare outcomes for animals [2]. The majority of studies to date have focused on non-human primates, and some of the variability in primate responses to visitors appears to be due to the characteristics of the habitats, such as the availability of visual barriers or retreat space [3,4,5,6], or characteristics of the visiting crowds, such as the number of people or how they behave [7,8,9,10,11].

In addition to the stress-related behavioral changes in individuals that may come as a response to visitor presence or activity, there is also evidence that visitors can incite increased social tension and aggression in primate social groups. While this aggression is often expressed within the social group itself (increased agonism among conspecifics), there are also examples of primates directing aggression at the visitors themselves. In open-contact situations, such as with free-ranging macaques in close proximity to tourists, primates often directly engage in aggressive interactions with nearby humans [12]. Even in a more traditional zoo setting in which the primates are separated from visitors by a barrier, primates have been reported to react to zoo visitors with directed aggression, such as displays. Mitchell et al. found that larger crowd sizes tended to increase the frequency of aggressive displays directed to the visitors [13] and Clark et al. observed that higher levels of environmental noise resulted in more instances of gorillas displaying aggression towards zoo visitors [14]. Such aggression may be targeted behavior in direct response to a perceived threat from humans, or it may be a form of redirected aggression in which a victimized individual expresses aggression to a third party. Redirected aggression is relatively widespread among primate species [15,16,17,18,19,20] and is often considered a conflict management strategy within the group. We are not aware of any focused empirical evaluation aimed at quantifying the factors that influence aggressive visitor–primate interactions in a zoo setting.

While most research has focused on how the characteristics of the visitor crowds affect the animals, it is also becoming apparent that characteristics of the animals themselves, such as their age or sex, play a role in explaining their reactions to zoo visitors [11,13,21]. There is growing interest in identifying additional factors given the amount of individual variation that has been observed [11,22,23,24]. One key dimension on which most individual primates vary within any given social group is social rank [25,26,27], yet the impact of rank on primate reactions to zoo visitors has not yet been explored. We focus here on one of the most despotic species, Japanese macaques (*Macaca fuscata*). Japanese macaque societies are characterized by steep, linear hierarchies in which rank has a strong influence on behavioral patterns [28]. Conflicts are primarily unidirectional, aggression is common, and reconciliation is rare [29]. Given the pervasive influence of rank in this species, we determined whether this factor influenced responses to visitors in a zoo setting. Unlike other individual differences, such as age or sex, an individual’s rank is a dynamic, unfixed quality based on the individual and the composition of the group [30].

Prior to beginning this study, we anecdotally observed periodic instances of macaques responding aggressively towards zoo visitors. The impetus for this investigation was to understand what factors, stemming from the zoo environment and from the animals themselves, may influence these interactions. Given previous research, demonstrating the influence of crowd size [22,31] and, separately, crowd activity levels [7,8] on animal behavior, we hypothesized that such factors would predict visitor-directed aggression. Additionally, we hypothesized that an individual’s social rank would also predict such behaviors. An enhanced understanding of the many types of factors that impact behavior may help influence future care and management of this species in zoological settings.

## 2. Materials and Methods

### 2.1. Subjects and Housing

The subjects of this study were 12 Japanese macaques in a social group at Lincoln Park Zoo’s Regenstein Macaque Forest. The group consisted of five adult females, three adult males, one juvenile male (who was transferred to another Association of Zoos and Aquariums (AZA)-accredited facility partway during the study), and three juvenile females. The large (685 m^2^) naturalistic outdoor exhibit was comprised of natural and artificial trees, bushes, large rocks, water features, grass, and mulch. There were two large viewing windows that the public could access (North viewing window: 10.5 m long by 1.9 m high; South viewing window: 14.8 m long by 2.7 m high; both starting 0.58 m off the ground). The two windows ran on only one side of the exhibit, providing macaques the ability to retreat from view (Figure 1 and Figure 2). There was a third, smaller viewing window (4.5 m long by 1.9 m high, starting 0.8 m off the ground) accessible from a locked meeting room used approximately once per week for small (approximately 10-person) staff and donor meetings; data were not collected at this window. The troop had access to additional indoor space out of public view during low temperatures or inclement weather. Daily, the group was given fresh produce and monkey chow scattered throughout their exhibit, access to environmental enrichment devices (e.g., puzzle feeders, mirrors, balls), and had access to water ad libitum. In addition, cognitive touchscreen studies were conducted with the troop on most weekdays [32].

### 2.2. Behavioral Data Collection

Behavioral data were collected by trained observers using the ZooMonitor software [33,34] on handheld tablet devices in the North and/or South viewing shelters along the East side of the habitat (Figure 1). Ten-minute observation sessions were conducted several times per day on most weekdays, and were nearly equally distributed between 10:00 and 16:00 from January 2017 to January 2018. Immediately prior to the start of each observation session, observers recorded crowd size and crowd activity levels. Crowd size was recorded categorically in ranges of ten (e.g., 0–9, 10–19, 20–29, etc.). Crowd activity was recorded categorically as calm, moderate, or frenetic, and reflected the observer’s subjective assessment of the volume and movement of the crowd independent of the number of people present (e.g., if several people were standing quietly, crowd activity was recorded as calm; if a few people were running around the exhibit and yelling, crowd activity was recorded as frenetic).

The behavioral data used for this project were collected as part of an ongoing behavioral research program in which focal individuals were pre-determined via a randomized schedule, resulting in near equal sampling of all subjects. Data for the ongoing behavioral research program were collected using 10-minute focal follows with instantaneous scan samples every 60 seconds, and select behaviors recorded on an all-occurrence basis for all troop members. All behaviors analyzed in this study fell into the latter category. Specifically, behaviors of interest for this study were non-contact aggression, contact aggression, and visitor-directed behavior. Non-contact aggression included lunging, rushing, chasing, and/or threats (head bob, threat bark, open mouth facial threat, ear flattening, brow-raising, ear-erecting), but did not involve physical contact. Contact aggression involved physical contact between individuals and included wrestling, lunging, hitting, slapping, pushing, grabbing, biting, and scratching. For any instance of non-contact and/or contact aggression during an observation (even if it did not involve the focal animal), the aggressor(s) and recipient(s) were recorded. In the case where one or both identities were not clear to the observer, the individual was noted as unknown. Visitor-directed behavior was categorized as threatening or non-threatening. Threatening visitor-directed behavior was defined as any aggressive behavior directed toward a visitor, including, but not limited to, slapping the glass, open mouth facial threatening, and/or extensive brow-raising. Non-threatening visitor-directed behavior included playful interactions and gentle physical contact with the visitor glass. Data were collected by trained observers who previously established inter-observer reliability of >85% agreement compared to an experienced researcher [35]. The macaques were out of view when at the meeting room viewing window, thus behaviors occurring at that window were not included in this study. This study was approved by the Lincoln Park Zoo Research Committee (2015-002), the governing body for all research conducted at this institution.

### 2.3. Calculation of Rank via Elo-Rating Scores

Japanese macaque ranks were calculated based on the Elo-rating approach that continuously updates rank based on recent interactions [36]. Generally speaking, Elo-ratings contain more information than classic cardinal rank orders and produce scores that indicate the relative distance between individuals. Elo-rating scores were established based on the all-occurrence instances of directional, dyadic non-contact aggression (because contact aggression was too rare, see below) as described above. We excluded cases in which there were multiple aggressors (N = 55) and/or multiple recipients (N = 50) of aggression in a single event, and cases when any identities associated with the event were unknown (N = 61), given lack of clarity about who was aggressing toward whom. Following the standard Elo-rating approach, individuals began with a score of 1000 that was updated as interactions (non-contact aggression) occurred, based on the following formula [36]:

Higher-rated individual wins:$${\mathrm{WinnerRating}}_{\mathrm{new}}={\;\mathrm{WinnerRating}}_{\mathrm{old}}+\left(1-p\right)\times k$$
$${\mathrm{LoserRating}}_{\mathrm{new}}={\;\mathrm{LoserRating}}_{\mathrm{old}}-\left(1-p\right)\times k$$

Lower-rated individuals wins (against the expectation):$${\mathrm{WinnerRating}}_{\mathrm{new}}={\;\mathrm{WinnerRating}}_{\mathrm{old}}+p\times k$$
$${\mathrm{LoserRating}}_{\mathrm{new}}={\;\mathrm{LoserRating}}_{\mathrm{old}}-p\times k$$

In the above formulas, *p* is the expectation of the higher-rated individual aggressing, which is a function of the absolute difference in the ratings of the two interaction partners before the interaction. *k* is a constant and determines the number of rating points that an individual gains or loses after a single encounter. For the purpose of this study, *k* was set at 100. At the beginning, *k* largely influences the rate at which Elo-ratings increase and decrease, but over time the influence of *k* diminishes, with a minor impact on the ultimate rankings [36].

Following this formula, the aggressor (expressed as “winner” in the formula) gains points over the target of their aggression, while the recipient (expressed as “loser” in the formula) loses points [37]. The amount of points gained or lost depends on the existing rank relationship between the recipient and aggressor and the probability of the observed outcome based on past interactions. Following an event, the change in points may be marginal or significant depending on the individuals involved and their history [36]. For example, if a male with a high Elo-rating aggresses towards a low ranking female, the increase in his score and the decrease in her score will be minimal. However, if the direction of the aggression was reversed, the change in both the aggressor and the recipient’s scores would be much greater. We retroactively extracted weekly Elo-ratings for all subjects for the duration of the observation period (52 weeks). All analyses were conducted in R [38], using the package EloRating [39].

### 2.4. Statistical Analysis

To determine whether rank, crowd size, and crowd activity predicted visitor-directed behavior in Japanese macaques, we created a mixed-effects logistic regression model that included Elo-rating nested in subjects as a random factor, crowd activity as a categorical fixed factor, and crowd size as a continuous fixed factor. Each subject had a constant Elo-rating value for all observation sessions that took place within the same week. Crowd activity and crowd size values were specific to each session, and the value for crowd size used in the analyses was the mid-point of the range selected (e.g., the value 5 was assigned if the observer had selected 0–9 at the start of the session). The outcome variable was visitor-directed behavior, scored as a binary variable per session to further ensure independence (e.g., for each individual, visitor-directed behavior was considered as present or absent each session, regardless of the frequency or duration of the behavior). Models did not include interaction terms since we had no a priori hypotheses related to the interactions. The visual inspection of residual plots did not reveal any obvious deviations from homoscedasticity or normality. We compared the Akaike information criterion with correction (AICc) values to compare model fits [40]. We also obtained *p*-values through likelihood ratio tests of the full model with the effect in question against the model without the effect in question, using the ANOVA function and Chi-square distribution. Tukey pairwise comparisons were used to compare levels of fixed effects remaining in the best fit model. All analyses were conducted in R, using the packages lme4 [41], AICcmodavg [42], and multcomp [43].

## 3. Results

Data were collected for 52 weeks for a total of 443.8 hours of observation with over 2663 sessions. A total of 1502 aggressive events between macaques with a single identified actor and recipient were recorded. Non-contact aggression was far more frequent (N = 1150, 76.6% of aggressive events), so we relied on the recurrent observations of non-contact aggression to calculate weekly Elo-ratings. Visitor-directed behaviors were more often threatening than non-threatening (total number of visitor-directed behaviors observed = 1574; 94.2% were threatening). Further examining the small subset of non-threatening visitor-directed behavior, revealed this behavior was primarily expressed by the juveniles, accounting for 83.7% of the visitor-directed behavior coded as non-threatening. Therefore we opted to focus our analyses on visitor-directed behavior categorized as threatening, hereafter referred to as visitor-directed aggression. Visitor-directed aggression occurred at least once in 23.3% of sessions and was expressed at least once by every macaque (median number of visitor-directed aggressive events across all macaques = 48, range 11 to 335).

The AICc values of the models revealed that the best model included both rank and crowd activity, but not crowd size (Table 1). Furthermore, likelihood ratio tests revealed that rank was the only significant predictor (Table 2). Considering these results together, it seems that rank has a strong relationship with visitor directed aggression, whereas crowd activity has a relationship but is less strong, with crowd size having no measurable relationship. The full results of the best model are provided in Table 3. Individuals holding lower rank positions were more likely to show visitor-directed aggression than higher-ranked individuals, and Tukey follow-up comparisons revealed that the activity level “frenetic” was associated with significantly more visitor-directed aggression than “calm” (*p* = 0.020; moderate vs calm, *p* = 0.690; moderate vs frenetic, *p* = 0.065, Figure 3).

## 4. Discussion

The effect of zoo visitors on animal behavior has been studied in depth, however, to our knowledge, this is one of the first evaluations of the factors that influence a particular type of response to visitors, visitor-directed aggression. We measured how characteristics of the visiting crowd, as well as the social rank of the individual, a variable that has rarely been considered, relate to visitor-directed aggression in a rather despotic species, the Japanese macaque. We find that social rank is indeed a strong predictor of visitor-directed aggression, and holds more explanatory power than both crowd activity level and crowd size.

Displaced aggression toward safe targets is one possible explanation for the pattern of results seen here. Japanese macaques are highly despotic, characterized by steep, linear hierarchies, and frequent high levels of aggression directed down the social hierarchy [44]. Third-party displacement aggression accounts for a large amount of aggression in some Old World primate species with similarly linear hierarchies [45,46,47]. For example, in savanna baboons (*Papio cynocephalus*) it is common to see a middle-ranking male receive aggression, then quickly turn and aggressively pursue a lower-ranking male who was not involved, who may then aggressively pursue a female who holds an even lower rank and was also not involved in the previous interactions [46]. Given the largely uni-directional route of aggression in Japanese macaque societies [19,44], higher-ranking individuals have more options for whom to direct aggressive behaviors toward without fearing retaliation. Lower-ranking individuals have increasingly fewer potential recipients to aggress toward and the public may provide a safe outlet for their aggression. This may be especially likely in our study troop since there are several meters of visitor glass where visitors are at a height nearly eye-to-eye with the macaques; visitors are an easy to locate and readily accessible target.

Another potential explanation for this pattern of rank-related results is that the lower-ranking macaques experience the public as more threatening than higher-ranking macaques do, perhaps due to their history of being targets of aggression, and they respond negatively toward the public due to this negative perception. However, considering the layout of the habitat at Lincoln Park Zoo makes this explanation seem unlikely. Although the visitor glass is expansive, it does not surround the macaques (Figure 1). Further, the habitat contains numerous trees that extend several meters above the height of the visitor glass as well as several square meters of natural and artificial landscaping. If the layout was different and the macaques were constrained to be in proximity to visitors, this explanation would seem more likely. However, given the opportunities the monkeys have to physically avoid being near visitors or in their view, it seems unlikely that the lower-ranking monkeys are experiencing high levels of perceived visitor threat.

Distinguishing between these two explanations is essential for determining whether visitor-directed aggression is an indicator of compromised welfare. If displaced aggression is the cause for lower-ranking individuals’ aggression to the public, then, in a sense, the public may be enhancing welfare by allowing a safe outlet for this behavior. However, if lower-ranking individuals are responding to perceived threats, this would suggest the public may be compromising welfare. One way to test whether the first proposed explanation, displaced aggression, accounts for the pattern of results observed here, would be to examine events occurring within the group preceding bouts of visitor-directed aggression. If aggressive events between conspecifics are taking place immediately before the individual aggresses toward the public, this would indicate evidence for redirection as seen in Aureli et al. [19].

Along with the rank-related results discussed above, our models indicated a significant increase in the frequency of visitor-directed aggression corresponding with frenetic crowd activity, but no relation with crowd size. Previous work has also found that crowd activity is the more meaningful dimension to consider [48]. While it seems reasonable that the macaques are responding to the activity levels, we cannot assume this singular direction of causality [8]. There may be a circular influence, or positive feedback loop, relating visitor-directed aggression and active crowds. For example, animals may be active and engaging in interesting behaviors independently of crowd characteristics, perhaps due to events within the group, causing visitors observing the active animals to become excited, increasing visitor activity. Given that our observation sessions often occurred in close succession, the crowd activity recorded at the start of a session could have been influenced by the macaque behavior in the previous session. Simply put, the animals may influence the guests’ behavior and vice versa. In future work we can disentangle this relationship and improve our design through a different sampling scheme that follows visitors and animals in real time.

In the future, it would be interesting to compare the amount of visitor-directed aggression expressed by closely-related macaque species that differ in their hierarchy steepness. The genus Macaca presents a natural test bed for this hypothesis given the broad spectrum of social tolerance expressed across species. Some species show less regimented directionality in their aggression, therefore, rank-related differences in visitor-directed aggression would not be expected. If the species could be tested in comparable habitats, we would predict that rank differences in visitor-directed aggression would be more apparent in the despotic species of macaques, such as Japanese and rhesus macaques (*Macaca mulatta*), than a less despotic species of the same genus with more potential outlets of aggression, such as Tonkean (*Macaca tonkeana*) and crested macaques (*Macaca nigra*) [44].

The present study adds at least two additional considerations to the current visitor effects literature, focusing on a specific type of behavior shown by zoo-housed macaques (visitor-directed aggression) and attempting to explain it based on characteristics of the zoo environment as well as the characteristics stemming from the animals and their relationships (rank). Here we see that both matter when trying to understand macaque behavioral responses to the public. This work serves to highlight the myriad of factors that may influence how animals behave in the zoo environment.

## 5. Conclusions

We sought to characterize how the presence and activity level of zoo visitors was associated with visitor-directed aggression demonstrated by zoo-housed Japanese macaques, and if such effects were influenced by the individual rank of those monkeys. We found the activity level of visitors was more influential than the size of the crowd and that they monkey’s social rank (especially lower-ranked individuals) predicted the visitor-directed aggression demonstrated at the exhibit’s glass barriers. We posit that this behavior may be characterized as redirected aggression that lower-ranked individuals cannot safely express to other (higher-ranking) monkeys in the group. Such findings emphasize the importance of not only studying potential visitor effects in a zoo setting but understanding such influences in the context of other factors such as social hierarchies. 

## Figures and Tables

**Figure 1 animals-09-00316-f001:**
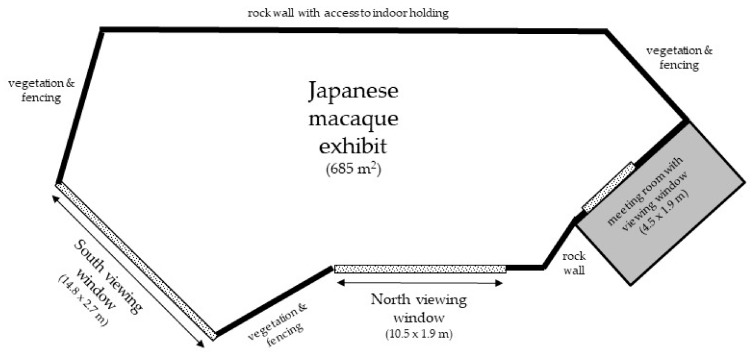
Japanese macaque exhibit showing the location of the viewing windows.

**Figure 2 animals-09-00316-f002:**
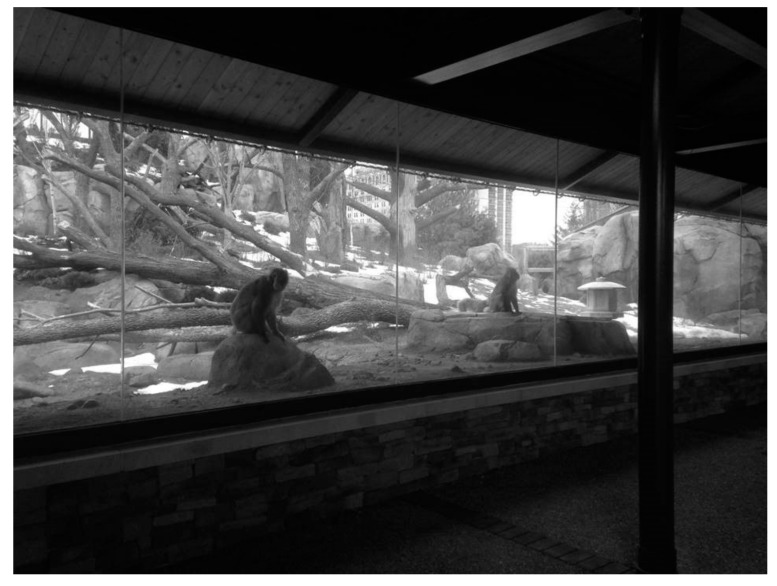
Section of the North viewing window of the Japanese macaque exhibit.

**Figure 3 animals-09-00316-f003:**
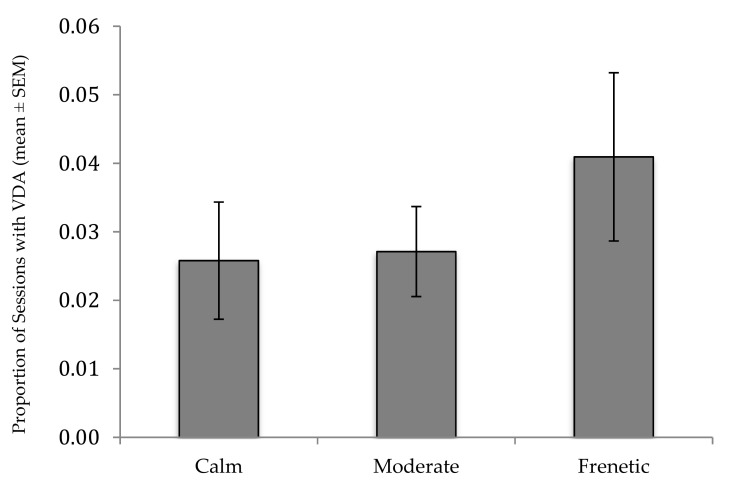
Proportion of sessions in which macaques displayed visitor-directed aggression by visitor activity level. Averages were calculated based on proportions calculated per subject; standard errors reflect variance across subjects.

**Table 1 animals-09-00316-t001:** Akaike information criterion with correction (AICc) values of mixed effects logistic regression models predicting visitor-directed aggression.

Factors Included in Model	AICc	Model Rank
Rank + Crowd Activity	7072.6	1 (best)
Rank + Crowd Size + Crowd Activity	7074.4	2
Rank	7075.0	3
Rank + Crowd Size	7075.9	4
Crowd Activity	7103.4	5
Crowd Activity + Crowd Size	7103.9	6
Crowd Size	7106.7	7 (worst)

**Table 2 animals-09-00316-t002:** Results of likelihood ratio tests comparing null models excluding factors of interest to the full model including rank, crowd size, and crowd activity.

Null Model	χ^2^	*p*-Value
Model excluding RANK	29.42	<0.0001
Model excluding CROWD SIZE	0.18	0.6755
Model excluding CROWD ACTIVITY	5.43	0.0663

**Table 3 animals-09-00316-t003:** Mixed effects logistic regression model results for best fitting model predicting the occurrence of visitor-directed aggression. The best-fitting model included crowd activity and rank, but did not include crowd size. The reference level for crowd activity was calm.

Fixed Factors	Beta	Lower-95	Upper-95	Std. Error
Intercept	−4.313	−4.495	−4.130	0.093
Crowd Activity (Frenetic)	0.552	0.143	0.960	0.209
Crowd Activity (Moderate)	0.075	−0.109	0.258	0.094
**Random Factor**	**SD**			
Rank	1.272

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
