# Peer review of "The Social Rank of Zoo-Housed Japanese Macaques is a Predictor of Visitor-Directed Aggression"

_animals, 2019, doi:10.3390/ani9060316_

Round 1
Reviewer 1 Report
This is a report of a study on the effect of zoo variables (visitor number and activity) and animal variables (rank) on the incidence of visitor-directed aggression in a zoo group of Japanese macaques. Although there have now been a number of studies on visitor effects on zoo animal behaviour, few have looked at human-directed behaviours, and none have looked at animal rank. Therefore this paper is a valuable addition to our understanding of visitor effects.
The paper is well written, but has several typographical errors (eg lines 78, 82, 124, 183), so needs final proof reading and correction if accepted.
The Methods are generally well explained, but I was a bit confused by the statement on animal identities in line 137, given that I thought (line 126-127) the animals were focal follows. If they weren’t focal follows (line 130), then why is focal via a randomised schedule mentioned at all? Maybe this is my failure of understanding, but this part of the Methods isn’t clear to me.
Otherwise a good, interesting paper.
Author Response
Dear Editor,
Thank you for considering our manuscript, “The social rank of zoo-housed Japanese macaques is a predictor of visitor-directed aggression”. We have revised our manuscript based on the helpful feedback provided by the two reviewers, with the changes documented in detail below. All line numbers refer to the revised and highlighted version. We hope you agree that the manuscript is improved, and look forward to hearing from you.
Sincerely,
Jocelyn M. Woods
Stephen R. Ross
Katherine A. Cronin
Reviewer 1:
1. This is a report of a study on the effect of zoo variables (visitor number and activity) and animal variables (rank) on the incidence of visitor-directed aggression in a zoo group of Japanese macaques. Although there have now been a number of studies on visitor effects on zoo animal behaviour, few have looked at human-directed behaviours, and none have looked at animal rank. Therefore this paper is a valuable addition to our understanding of visitor effects.
a. Thank you.
2. The paper is well written, but has several typographical errors (eg lines 78, 82, 124, 183), so needs final proof reading and correction if accepted.
a. All the listed errors have been corrected, and we have read through for remaining typographical errors and corrected those as well.
3. The Methods are generally well explained, but I was a bit confused by the statement on animal identities in line 137, given that I thought (line 126-127) the animals were focal follows. If they weren’t focal follows (line 130), then why is focal via a randomised schedule mentioned at all? Maybe this is my failure of understanding, but this part of the Methods isn’t clear to me.
a. Thanks to this comment and a similar comments provided by Reviewer 2, we realize that there was some clarification needed in our methods specifically pertaining to the difference between the data used to answer the question posed for this project and the data collected for the ongoing behavioral research program. We hope that edits made in lines 126, 128, and 136-138 help clarify.
4. Otherwise a good, interesting paper.
a. Thank you.
Reviewer 2 Report
This is overall an interesting paper. However I have a few concerns as follows:
Title: I believe it would be helpful to add ‘social’ before ‘rank’ in the title to make it clearer.
There is at least a couple of places where abbreviations are used and not explained.
Simple summary: I think that the results indicate something rather than using the past tense (because the results still indicate the same thing). The actual research to obtain the finding is in the past tense. I think line 18-9 should read “The results indicate that the social rank of a monkey is a significant predictor of …” There are similar example to this throughout.
2.2 Behavioral data collection: The data were collected at random times over a long period to time. We need more information about these collection sessions. Were all times of the day covered equally and if not why not. We also need to know more about the technique that was used, e.g. were the cameras (tablets) held still focused on one spot of the enclosure, follow one individual or did they sweep across the enclosure for the ten minutes?
It says that the crowd was assessed prior to recording for ten minutes but vast changes in the crowd can occur in ten minutes. Noisy kids can come and go and large groups can arrive and leave easily within ten minutes. Therefore it is difficult to link behaviour to this ever changing visitor profile. How was this handled?
I find the second paragraph very confusing particularly how it ties in with the data collection details provided in the first paragraph. Could you please clarify. If this is clarified then I think the next section (2.3) will become clearer.
Line 205: The number of sessions and hours of observation don’t seem to match up. 2663 sessions is equal to 26630 minutes (according to the 10 minute sessions mention in methodology) and this is 443.8 hours. This needs to be clarified.
Discussion: In a couple of places it states that glass extends along one wall (southern wall presumably). The discussion ignores the second stretch of glass (though smaller) on the northern wall).
Author Response
Dear Editor,
Thank you for considering our manuscript, “The social rank of zoo-housed Japanese macaques is a predictor of visitor-directed aggression”. We have revised our manuscript based on the helpful feedback provided by the two reviewers, with the changes documented in detail below. All line numbers refer to the revised and highlighted version. We hope you agree that the manuscript is improved, and look forward to hearing from you.
Sincerely,
Jocelyn M. Woods
Stephen R. Ross
Katherine A. Cronin
Reviewer 2:
This is overall an interesting paper. However I have a few concerns as follows:
1. Title: I believe it would be helpful to add ‘social’ before ‘rank’ in the title to make it clearer
a. We agree. Revision has been made.
2. There is at least a couple of places where abbreviations are used and not explained.
a. All abbreviations have been spelled out at their first mention, with the exception of GLMM mentioned in the abstract due to word count limitations (line 34).
3. Simple summary: I think that the results indicate something rather than using the past tense (because the results still indicate the same thing). The actual research to obtain the finding is in the past tense. I think line 18-9 should read “The results indicate that the social rank of a monkey is a significant predictor of …” There are similar example to this throughout.
a. We agree and have made revisions to the line indicated as well as throughout the text to ensure that results are in the past tense and implications are in present or future tense.
4. 2.2 Behavioral data collection: The data were collected at random times over a long period to time. We need more information about these collection sessions. Were all times of the day covered equally and if not why not. We also need to know more about the technique that was used, e.g. were the cameras (tablets) held still focused on one spot of the enclosure, follow one individual or did they sweep across the enclosure for the ten minutes?
a. We recognize our methods needed further clarification and we made edits to improve clarity. See lines 116-119 for information on the time of day observations were taken and the near equal sampling throughout the day. In these lines we also clarified our behavioral data collection process in which observers used handheld tablets to manually record behaviors, not through video. We hope these clarifications make our methods more clear.
5. It says that the crowd was assessed prior to recording for ten minutes but vast changes in the crowd can occur in ten minutes. Noisy kids can come and go and large groups can arrive and leave easily within ten minutes. Therefore it is difficult to link behaviour to this ever changing visitor profile. How was this handled?
a. We acknowledge this may be a limitation of our study which is now indicated in line 294-295. However this approach has been used in previous studies examining visitor effects (e.g., Bonnie, Ang & Ross, 2016, Effects of crowd size on exhibit use by and behavior of chimpanzees (Pan troglodytes) and Western lowland gorillas (Gorilla gorilla) at a zoo, Applied Animal Behaviour Science, 178, 102-110) giving us confidence that while not an exact, real-time measure, the relationship does not change so dramatically over a ten-minute period that patterns are not detectable.
6. I find the second paragraph very confusing particularly how it ties in with the data collection details provided in the first paragraph. Could you please clarify. If this is clarified then I think the next section (2.3) will become clearer.
a. Similar to Reviewer 1’s comment, we revised some statements in order to make clarifications between the subset of data used for this project and the data recorded as a part of the ongoing behavioral research program. See lines 126 and 128.
7. Line 205: The number of sessions and hours of observation don’t seem to match up. 2663 sessions is equal to 26630 minutes (according to the 10 minute sessions mention in methodology) and this is 443.8 hours. This needs to be clarified.
a. Thank you for calling this to our attention, it has been corrected.
8. Discussion: In a couple of places it states that glass extends along one wall (southern wall presumably). The discussion ignores the second stretch of glass (though smaller) on the northern wall).
a. Thank you for calling attention to this, we have now clarified. Specifically, revisions have been made in lines 259-262 and 267-270 to clarify that observations were taken on the only side of the exhibit (the East side) where the visitor glass was present. The two sections of visitor glass along the east side are labeled as South and North viewing windows in Figure 1.
Round 2
Reviewer 2 Report
Thank you for addressing my concerns.